# Exercise and Nutrition for Sarcopenia: A Systematic Review and Meta-Analysis with Subgroup Analysis by Population Characteristics

**DOI:** 10.3390/nu17142342

**Published:** 2025-07-17

**Authors:** Yong Yang, Neng Pan, Jiedan Luo, Yufei Liu, Zbigniew Ossowski

**Affiliations:** 1School of Sports Training, Chengdu Sports University, Chengdu 641418, China; yong.yang@awf.gda.pl; 2Department of Physical Culture, Gdansk University of Physical Education and Sport, 80-336 Gdansk, Poland; neng.pan@awf.gda.pl (N.P.); jiedan.luo@awf.gda.pl (J.L.); yufei.liu@awf.gda.pl (Y.L.)

**Keywords:** sarcopenia, exercise, nutrition, older adults, combined intervention

## Abstract

**Background:** Sarcopenia significantly affects the health and quality of life in older adults. Exercise combined with nutritional interventions is widely recognized as an effective strategy for improving sarcopenia outcomes. However, current studies rarely focus on differential effects across subpopulations with distinct demographic and health characteristics. This study aimed to explore the effects of combined exercise and nutrition interventions on sarcopenia-related outcomes, considering the variations in population characteristics. **Methods:** A systematic search was conducted across PubMed, Embase, the Web of Science, and Cochrane Library, covering the literature published between January 2010 and March 2025. Only randomized controlled trials (RCTs) evaluating combined exercise and nutritional interventions for sarcopenia were included. The primary outcomes were handgrip strength (HS), the skeletal muscle mass index (SMI), gait speed (GS), and the five-times sit-to-stand test (5STS). The mean differences (MD) with 95% confidence intervals (CIs) were calculated. Random-effects models were used for the meta-analysis and subgroup comparisons. **Results:** Fifteen RCTs involving 1258 participants in the intervention group and 1233 in the control group were included. Exercise combined with nutritional interventions significantly improved sarcopenia-related outcomes. HS improved with a pooled MD of 1.77 kg (95% CI: 0.51 to 3.03, *p* = 0.006); SMI increased by 0.22 kg/m^2^ (95% CI: 0.09 to 0.35, *p* = 0.0007); GS improved by 0.09 m/s (95% CI: 0.04 to 0.14, *p* = 0.0002); and 5STS performance improved with a time reduction of −1.38 s (95% CI: −2.47 to −0.28, *p* = 0.01). Subgroup analyses indicated that the intervention effects varied according to age, BMI, and living environment. **Conclusions:** Exercise combined with nutrition is effective in improving key outcomes associated with sarcopenia in older adults. The magnitude of these effects differed across population subgroups, underscoring the importance of tailoring interventions to specific demographic and health profiles.

## 1. Introduction

Sarcopenia is a systemic disorder characterized by a progressive loss of skeletal muscle mass, strength, and function, with a significantly increased risk of falls, dysfunction, weakness, and death [1,2]. Its pathogenesis is not fully understood and may be closely related to multiple factors, such as age-related hormonal changes (e.g., testosterone and estrogen), chronic diseases (e.g., diabetes mellitus and hypertension), nutritional deficiencies (e.g., protein and micronutrient deficiencies), and physical inactivity [3,4,5,6,7]. Epidemiological studies have shown that the prevalence of sarcopenia is higher in postmenopausal women and the elderly population [8,9,10,11], with prevalence rates ranging from 8% to −36% in those under 60 years of age, 5–13% in those aged 60–70 years, and as high as 11–50% in those over 80 years of age [12,13]. This poses a serious challenge to the quality of life of the elderly and the public health system, and effective interventions are urgently needed to improve muscle function and reduce the risk of falls and disabilities.

Currently, exercise and nutritional interventions are considered the most promising non-pharmacological treatments for alleviating sarcopenia [14,15]. Leading sarcopenia guidelines (e.g., EWGSOP2, AWGS 2019) endorse exercise as the primary intervention [16,17]. Exercise improves muscle metabolism through multiple pathways [18]. Nutritional interventions, as important complements to exercise interventions, also play a key role in sarcopenia management. High-quality protein, vitamin D, and omega-3 (ω-3) fatty acids are the most widely used nutrients. A daily intake of 1.3–1.6 g/kg body weight of high-quality protein can delay muscle loss and enhance functional status [19,20]. Among these, leucine-rich animal and whey proteins have unique advantages in stimulating the mTOR pathway [21]. Vitamin D regulates calcium channels through the vitamin D receptor (VDR), enhances muscle contraction, and maintains the calcium–phosphorus balance. ω-3 fatty acids can improve the muscle metabolic environment by inhibiting the NF-κB pathway and lowering the levels of inflammatory factors [21,22,23]; however, there is a lack of clear evidence on its optimal dosage, safety, and intervention period in different populations.

Meta-analyses have been conducted to explore the overall intervention effects of exercise combined with nutritional interventions in patients with sarcopenia [24,25,26,27,28,29]. Most studies have focused on the differences in effects between combined and single interventions, with fewer systematic, hierarchical, and comprehensive assessments of different population characteristics and intervention strategy variables. These population characteristics not only influence the physiological response of patients to interventions but may also determine the optimal combination of intervention programs and intervention intensity.

Although interventions combining exercise and nutrition are widely recognized as effective strategies for improving sarcopenia, existing research often ignores the wide variation that exists among older adults. Clinical presentation and response to interventions vary widely according to age, body mass index (BMI), and living environment (e.g., community versus institutional care) [24,30,31]. Treating older adults as a homogeneous population may mask specific differences between these subgroups and limit the generalizability of the findings. Given the heterogeneity of sarcopenia across populations, subgroup analyses are essential to determine the true efficacy of comprehensive interventions. Prior evidence suggests significant differences in clinical presentation and response to intervention by age, body mass index status, and living environment (e.g., community-dwelling versus institutionalized) [31,32,33]. In addition, individual differences in adherence and physiological responses to exercise or nutritional interventions are evident in specific subgroups, such as underweight, advanced age, or care-dependent populations. However, only a few meta-analyses have systematically explored these differences. Therefore, our study aimed to fill this gap by stratifying analyses of key population variables to provide more targeted evidence for personalized intervention strategies in the management of sarcopenia.

## 2. Materials and Methods

### 2.1. Protocols and Registration

This systematic review and meta-analysis was conducted and reported in accordance with the Preferred Reporting Items for Systematic Analyses [34], which ensure transparency, methodological rigor, and reproducibility in evidence synthesis. A detailed protocol outlining the objectives, eligibility criteria, search strategy, and planned methods of analysis was prospectively registered in the International Prospective Register of Systematic Reviews (PROSPERO; Registration No. CRD420251036334). Registration prevents outcome reporting bias and enhances the credibility and replicability of the research process.

### 2.2. Search Strategy

A comprehensive literature search was conducted to identify relevant English-language randomized controlled trials (RCTs) published between 1 January 2010, and 31 March 2025. This 15-year period was chosen to reflect both foundational developments and the most recent research progress in exercise and nutritional interventions for sarcopenia. The search was completed before the PROSPERO registration date (19 April 2025).

Four major electronic databases were searched: PubMed, Embase, the Web of Science, and Cochrane Library. The search strategy included controlled vocabulary and free-text terms relevant to sarcopenia, exercise, nutrition, aging populations, and randomized controlled trials. The complete search string was as follows: ((“sarcopenia” [Title/Abstract] OR “muscle wasting” [Title/Abstract] OR “muscle loss” [Title/Abstract] OR “age-related muscle atrophy” [Title/Abstract] OR “muscle atrophy” [Title/Abstract] OR “skeletal muscle depletion” [Title/Abstract] OR “frailty syndrome” [Title/Abstract] AND (“resistance training” [Title/Abstract] OR “aerobic exercise” [Title/Abstract] OR “physical activity” [Title/Abstract] OR “Nordic walking” [Title/Abstract] OR “strength training” [Title/Abstract] OR “weight training” [Title/Abstract] OR “progressive resistance exercise” [Title/Abstract] OR “combined exercise” [Title/Abstract] OR “endurance training” [Title/Abstract] OR “home-based exercise” [Title/Abstract] OR “supervised exercise” [Title/Abstract] OR “balance training” [Title/Abstract] OR “mobility training” [Title/Abstract] OR “exercise therapy” [Title/Abstract])AND(“nutrition” [Title/Abstract] OR “dietary supplements” [Title/Abstract] OR “protein supplementation” [Title/Abstract] OR “amino acids” [Title/Abstract] OR “leucine” [Title/Abstract] OR “HMB” [Title/Abstract] OR “whey protein” [Title/Abstract] OR “BCAA” [Title/Abstract] OR “vitamin D” [Title/Abstract] OR “omega-3 fatty acids” [Title/Abstract] OR “nutritional intervention” [Title/Abstract]) AND (“older adults” [Title/Abstract] OR “elderly” [Title/Abstract] OR “aging” [Title/Abstract] OR “community-dwelling” [Title/Abstract] OR “frail elderly” [Title/Abstract] OR “postmenopausal women” [Title/Abstract]) AND (“randomized controlled trial” [Publication Type] OR “RCT” [Title/Abstract] OR “clinical trial” [Publication Type]) AND (“1 January 2010” [Date—Publication]: “31 March 2025” [Date—Publication]).

To enhance the comprehensiveness of our search and minimize publication bias, we also screened gray literature sources, such as Google Scholar and Open Grey, and selected conference proceedings using the same keywords. However, none of the identified records met the eligibility criteria due to the lack of randomization, insufficient reporting quality, or unclear interventions and were therefore excluded. We also manually reviewed the reference lists of all the included studies and relevant reviews to ensure that no eligible studies were missed.

### 2.3. Eligibility Criteria

Our search strategy was based on the (PICOS) principle. Population (P): Older adults with a confirmed diagnosis of sarcopenia, including age-based subgroups (<65 years vs. ≥65 years), excluding older adults with disease, hypertension, hematologic disorders, and liver disease. Intervention (I): A combination of exercise and nutritional interventions. This study examined structured exercise programs (≥3 metabolic equivalents [METs]) used to manage sarcopenia, such as resistance training, aerobic training, or combined training. This study also included conventional nutritional support (e.g., protein, vitamin D, and amino acids) from the diet but excluded medications and sedentary activities. Comparison (C): Activities of Daily Living. Outcome (O): Changes in key indicators of sarcopenia, including handgrip strength (HS), skeletal muscle mass index (SMI), gait speed (GS), and 5-item sit-to-stand test (5STS). The study design (S) included randomized controlled trials (RCTs) for quantitative synthesis.

### 2.4. Screening Criteria for Studies

Two authors (Y.Y. and J.D.L.) independently checked the titles and abstracts of studies that met the inclusion criteria. If one of the authors felt that a research paper met the criteria, the full text of the paper was retrieved. Subsequently, both the authors independently assessed whether the entire article met these requirements. If an agreement could not be reached, a third author (N.P.) made the decision through debate.

### 2.5. Data Extraction

Two authors (Y.Y. and J.D.L.) independently extracted data for inclusion in the study. The primary outcomes were the skeletal muscle index (SMI), skeletal muscle mass (SMM), HS, GS and 5 STS. An Excel spreadsheet was designed a priori to extract relevant data, including publication characteristics (author name, country, and the year of publication), methodological characteristics (intervention and sample size), participant characteristics (age and sex), comprehensive nutritional exercise characteristics (intervention period, the frequency of intervention, and duration), risk assessment, and outcome characteristics.

In our data extraction efforts, we assessed the effects of combining exercise and nutrition on sarcopenia in various populations. Two authors (Y.Y. and J.D.L.) conducted cross-sectional assessments based on different exercise and nutrition interventions (exercise modality, frequency, intensity, and duration), and if any differences existed, a third author (N.P.) intervened until the differences were eliminated.

### 2.6. Risk of Bias Assessment

The risk of bias for each included study was independently assessed by two reviewers (Y.Y. and J.D.L.) using the Cochrane Collaboration Risk of Bias Assessment Tool [30]. In the event of a disagreement, a third reviewer mediated the discussion to reach consensus. This process ensured consistency and minimized subjective bias in the quality evaluation of the included randomized controlled trials. This was a randomized controlled trial. Assessment indicators included incomplete outcome data (attrition bias), reporting bias (reporting bias), personnel and participant blinding (performance bias), allocation concealment (selection bias), randomized sequence generation (selection bias), and other biases. The risk of bias was categorized into three categories: ‘low,’ ‘unclear’ and ‘high’.

### 2.7. Data Synthesis and Analysis

This review study used Review Manager (RevMan) 5.4.1 software to statistically summarize and analyze the included data. For different outcome indicators, models were selected according to the degree of heterogeneity: a fixed-effects model was used when *I*^2^ was <50% and a random-effects model when *I*^2^ was ≥50%. If different studies used different scales to measure the same indicator, the standardized mean difference (SMD) was used; if the scales were consistent, the mean difference (MD) was used for combined analysis.

Cohen’s kappa coefficient was used to assess the consistency between two independent reviewers to ensure the reliability of the data extraction and risk of bias assessment. The results ranged from 0.71 to 0.89, which indicated that the assessment results had good to excellent consistency and enhanced the robustness of data processing in this study.

According to the recommendations of the Cochrane Handbook of Systematic Evaluation (version 6.3), *I*^2^ > 50% was used as the threshold for judging substantial heterogeneity and moderate heterogeneity when *I*^2^ was between 30% and 60%. Thus, the criteria for heterogeneity were consistent with internationally accepted norms.

To further explore the differences in the effects of the combined intervention under different population characteristics, we conducted statistical tests on the results of each subgroup using the chi-square test (χ^2^) under the random-effects model to assess whether the differences between the subgroups were statistically significant and combined with the *I*^2^ indicator to judge the heterogeneity between the groups. A significant difference between subgroups was considered at *p* < 0.05. This analysis helped to reveal the moderating effect of factors such as age, BMI, living environment, and whether myasthenia gravis was diagnosed based on the effect of the intervention.

We also performed sensitivity and impact analyses to assess the impact of individual studies on combined outcomes. Despite the high heterogeneity of some of the outcomes (e.g., grip strength *I*^2^ = 83%), the overall direction of the effect was consistent, and the statistical significance was stable, suggesting the good robustness of the findings.

Subgroup analyses were predefined prior to the literature screening and stratified based on clinically relevant characteristics (e.g., age, BMI, myasthenia gravis diagnostic status, and residential setting) to explore the variability of the intervention effects in different populations and improve the explanatory power and clinical utility of the results.

In addition, we conducted further subgroup analyses based on differences in the intervention strategies themselves, including exercise mode, frequency (1–2, 3–4, ≥5 times per week), intervention duration (4–12 weeks, 13–24 weeks, >24 weeks), and the nutritional form of intake (e.g., daily protein intake <1.2 g/kg/d, 1.2–1.5 g/kg/d, >1.5 g/kg/d, or vitamin D intake <800 IU versus ≥800 IU).

Considering that the diagnostic criteria for sarcopenia used in different studies may have an impact on the outcome, we collated and summarized the diagnostic criteria used in each study: 6 studies from China (Liu-Ying Zh, Zhuo Li, Xinyi Liao), Japan (Minoru Yamada), South Korea (Sunghwan Ji), and Turkey (Huzeyfe Ancı) were more likely to use the same diagnostic criteria as those used in the other studies. Ancı studies mostly used the Asian Working Group on Myasthenia Gravis Standards 2019 (AWGS 2019) [35,36,37,38,39,40]; while 9 studies from the USA (Daniel Rooks, Caitlin Mason), Italy (Mariangela Rondanelli, Valentina Muollo), Switzerland (Anna K Eggimann), Austria (Eva M Strasser), Sweden (Sanna Vikberg), the United Kingdom (Antoneta Granic), and Denmark (Josephine Gade), the 2018 European Working Group on Geriatric Sarcopenia Revised Criteria (EWGSOP2) [41,42,43,44,45,46,47,48,49] were commonly used. These two types of diagnostic criteria have some differences in the assessment indicators and the setting of cutoff values, which may have an impact on the heterogeneity of the study results to some extent. We labeled the diagnostic criteria used in the studies in Table 1 to understand the characteristics of the studies in the context of the different criteria.

## 3. Results

### 3.1. Study Selection

The literature search included PubMed (705 articles), Embase (1346 articles), the Web of Science (1203 articles), and Cochrane Library (468 articles), totaling 3722 articles. Eight articles were retrieved by using other search methods. After removing duplicates (2641 articles), 1089 articles remained. After an initial review of titles and abstracts, 644 articles remained, which was followed by further screening, with 629 full-text exclusions, including the exclusion of non-randomized controlled studies (n = 323), the exclusion of incomplete data (n = 66), the exclusion of articles in sessions (n = 43), the exclusion of outcomes that were not eligible for inclusion in the present review (n = 66), and the exclusion of interventions that were not eligible for inclusion in the present review (n = 131), leaving 15 articles after the final assessment of the full text [35,36,37,38,39,40,41,42,43,44,45,46,47,48,49] (Figure 1).

### 3.2. Study Characteristics

Fifteen studies recruited people aged 66.47 years or older (1258 in the intervention group and 1233 in the control group). These studies covered the following outcome measures, the skeletal muscle index (SMI), hand grip strength (HS), gait speed (GS), and five-times sit-to-stand test (5STS), as detailed in Table 1. There were four articles that examined the outcome indicator “SMI”. There were 11 articles on the outcome indicator “HS”. There were seven articles on the outcome indicator “GS” and four articles on the outcome indicator “5 STS”, including articles from China, Sweden, Turkey, Switzerland, Austria, Denmark, Japan, Italy, the United States, Korea, and the United Kingdom.

To enhance the visualization of the evidence contribution, we added a flow diagram (Figure 2) in the Results Section. This diagram presents the number and distribution of studies contributing to each primary outcome measure, including handgrip strength (HS), the skeletal muscle mass index (SMI), gait speed (GS), and the five-times sit-to-stand test (5STS). This visual representation facilitates a clearer understanding of the evidence base supporting each outcome and allows readers to quickly grasp the scope and strength of the supporting data across the included studies. A brief explanation is provided in the figure captions below.

Tests for subgroup differences were conducted for each of the primary outcome measures (HS, SMI, GS, and 5STS) to determine whether the intervention effects varied significantly across participant characteristics. The results of these interaction tests, including χ^2^ and *I*^2^ values, are presented in the figure captions and the main text. Statistically significant subgroup effects were observed in several comparisons, supporting the relevance of population characteristics in modulating treatment responses.

The intervention periods included in the studies ranged from 6 weeks to 3 years, with interventions occurring 1–7 days per week and single interventions lasting 15–90 min. The intervention consisted of regular RT training and walking, including the administration of protein, milk, fat, vitamins, omege3 and fish oil. The intervention in the control group was daily living without physical activity (Table 1).

### 3.3. Risk of Bias

The results of the risk of bias assessment of the studies included in the systematic reviews and meta-analyses are presented in Figure 3. The risk of bias was assessed using the Cochrane Risk of Bias 2.0 (RoB 2) Tool [50], which evaluates bias across five domains relevant to randomized trials. Overall, 80–90% of the studies had a low risk of bias in randomized sequence generation and allocation concealment, and the results showed that the study designs of the 15 included studies generally adopted reasonable randomization methods, which adequately ensured the scientific validity and fairness of the trial groupings.

However, on the dimensions of the “blinding of participants and personnel” and “blinding of outcome assessment,” 50–60% of the studies were rated as having unclear risk, which indicated that most of the studies had omissions or methodological flaws in the blinding design, which might affect the objectivity of the study results.

In terms of incomplete outcome data, approximately 85% of the studies had a low risk, but 10% had a high risk, suggesting that individual studies had missing follow-up data, which may weaken the reliability of the conclusions.

In addition, the proportion at risk for selective reporting is unclear, suggesting that approximately 50% of the studies may not have fully reported the prespecified findings, increasing the likelihood of publication bias. In contrast, the overall low risk of “other bias” suggests that most studies did not identify substantial additional systematic bias.

In conclusion, the overall quality of the studies included in this systematic review was relatively manageable; however, some uncertainty remained regarding blinding implementation and the reporting of results. Caution must be exercised when interpreting the final conclusions and rationalized in the context of specific studies.

The methodological quality of the 15 articles included in this study, all of which were randomized controlled trials, was assessed using the Cochrane Collaboration’s risk of bias assessment tool (Figure 4). Regarding the generation of randomized sequences, 11 studies clearly described randomization methods (e.g., computerized randomization and randomized table of numbers), but 4 studies did not report specific allocation methods, which may be subject to selection bias. Due to the specific nature of exercise combined with nutritional interventions, it is difficult for studies to be completely double-blind. Only eight studies used blinding (e.g., blinding of assessors), and two studies did not specify whether the participants and researchers were blinded, which could lead to performance bias.

In terms of outcome assessment bias, four studies did not use blinding during data analysis, and one study explicitly stated that outcome assessors were not blinded, which may have affected the objectivity of indicators related to sarcopenia (e.g., muscle mass, grip strength, and step speed). In addition, three studies reported that subjects dropped out during the course of the experiment but did not adequately analyze whether the reason for dropping out was related to the intervention or severity of sarcopenia, which may have contributed to attrition bias. Five studies may have had selective reporting, such as the incomplete reporting of prespecified outcomes (e.g., reporting only improvements in muscle mass and omitting changes in functional indices). Other risks of bias in the remaining four studies were unclear and needed to be further assessed based on a specific experimental design.

### 3.4. Grip Strength Results and Subgroup Analysis

There were a total of 12 studies related to grip strength (HG). There were 421 and 427 participants in the exercise intervention and control groups, respectively. Overall, exercise combined with nutritional interventions had a significant intervention effect on HG metrics (MD, 1.38 [95% CI: 0.28, 2.49], *p* = 0.01), but there was a high degree of heterogeneity across studies (*I*^2^ = 83%). Therefore, statistical analyses were performed using a random-effects model. Mean difference (MD) was used as an effect size indicator. The studies were analyzed in subgroups based on the following four factors, body mass index, age, diagnostic subgroups, and living environment, as shown in Figure 5.

For subgroup comparisons of body mass index, BMI-Abnormal included six studies and BMI-Normal included six studies. As shown in Figure 5, a subgroup analysis was performed. In BMI-Abnormal, the MD was 1.75 (95% CI: −2.05, 5.56), *I*^2^ = 88%, *p* = 0.37. This suggests that exercise combined with nutrition did not significantly improve the BMI-Abnormal group and there was a high degree of heterogeneity among the studies. In the BMI-Normal group, the MD was 0.93 (95% CI:−1.89, 3.76), *I*^2^ = 76%, *p* = 0.52), suggesting that exercise combined with nutrition did not have a significant ameliorative effect on the BMI-Normal group and that there was a high degree of heterogeneity between studies.

In the age subgroup comparison, 11 studies involved people over 65 years old and 1 study involved people under 65 years old, as shown in Figure 5. In the subgroup aged ≥65 years, the MD was 1.03 (95% CI: −1.45, 3.51), *I*^2^ = 84%, *p* = 0.42), suggesting that exercise combined with nutritional interventions did not have a significant effect on improvement in the over 65s and there was a high degree of heterogeneity between studies. In the under-65 population, the MD was 4.67 (95% CI: 1.96, 7.38), *p* = 0.0007), suggesting that exercise significantly improved the under-60 population.

In the comparison of diagnostic subgroups, 1 study included suspected patients and 11 studies included confirmed patients, as shown in the subgroup analysis in Figure 5. In the suspected subgroup, the MD was −0.10 (95% CI: −10.99, 10.79), *p* = 0.99. This finding suggests that the effect of exercise combined with nutrition was not significant. In the confirmatory subgroup, the MD was 1.43 (95% CI: −0.95, 3.82), *I*^2^ = 86%, *p* = 0.24). This suggests that the intervention effect of exercise combined with nutrition was not significant, and that there was a high degree of heterogeneity between the studies.

In the residential setting subgroup comparisons, 11 studies involved independently housed older adults and 1 study involved older adults in care settings, as shown in the subgroup analyses in Figure 5. In the independently housed subgroup, the MD was 1.55 (95% CI: −1.07, 4.17), *I*^2^ = 85%, and *p* = 0.25. This suggests that the combination of exercise and nutritional interventions did not have a significant ameliorative effect on the group of independently housed older adults and that there was a high degree of heterogeneity among the studies. In the subgroup of older adults in care settings, the MD was −0.16 (95% CI: −2.32, 2), *p* = 0.88. This finding suggests that the combination of exercise and nutritional intervention did not have a significant ameliorative effect on older adults in care settings.

Subsequently, a publication bias test was performed using REVIEW MANAGER 5.4.1. As shown in Figure 6, the studies were symmetrically distributed; therefore, it can be judged that there was no publication bias.

### 3.5. SMI Results and Subgroup Analysis

Five studies involving the skeletal muscle mass index (SMI) were conducted. A total of 188 individuals were included in the exercise intervention group and 173 in the control group. Overall, exercise combined with nutritional interventions had a significant effect on SMI metrics (MD: 0.18 [95% CI: 0.17, 0.2], *p* < 0.00001), with low between-study heterogeneity (*I*^2^ = 29%). Therefore, a fixed-effects model was used for statistical analysis. The mean difference (MD) was used as an indicator of effect size. The studies were analyzed in subgroups based on the following four factors, body mass index, age, myasthenia gravis diagnosis, and living environment, as shown in Figure 7.

For subgroup comparisons of body mass index, the normal body mass index included two studies, and the normal body mass index included three studies. Subgroup analyses are shown in Figure 7. In BMI-Abnormal, the MD was 0.19 (95% CI: 0.17, 0.21), *I*^2^ = 0%, *p* < 0.00001. This indicates that the intervention of exercise combined with nutrition had a significant improvement effect on the BMI-Abnormal group, and there was no heterogeneity among the studies. In the BMI-Normal group, the MD was 0.08 (95% CI:−0.01, 0.18), *I*^2^ = 0%, *p* = 0.08, suggesting that the intervention of exercise combined with nutrition did not have a significant ameliorative effect on the BMI-Normal group.

In the comparison of age subgroups, four studies involved people over 65 years of age and one study involved people under 65 years of age. Subgroup analyses are shown in Figure 5. In the subgroup of ≥65 years old, the MD was 0.19 (95% CI: 0.17, 0.21), *I*^2^ = 0%, *p* < 0.00001. This suggests that exercise combined with nutrition interventions have a significant ameliorative effect on the over-65-years-old group and there was no heterogeneity among the studies. In the under-65 age group, the MD was 0.07 (95% CI: −0.03, 0.17), *p* = 0.16. This finding suggests that exercise combined with nutritional interventions did not have a significant ameliorative effect in the under-60 age group.

In the comparison of diagnostic subgroups, one study included suspected patients and four studies included confirmed patients. The subgroup analyses are shown in Figure 7. In the suspected subgroup, the MD was −0.1 (95% CI: −1.07, 0.87), *p* = 0.84). This suggests that the effect of exercise combined with nutrition was not significant. In the confirmatory subgroup, the MD was 0.18 (95% CI: 0.16, 0.21), *I*^2^ = 52%, *p* < 0.00001. This suggests that the intervention effect of exercise combined with nutrition was significant; however, there was a high degree of heterogeneity between studies.

Subsequently, a publication bias test was performed using REVIEW MANAGER 5.4.1. As shown in Figure 8, the studies were symmetrically distributed; therefore, it can be judged that there was no publication bias.

### 3.6. Gait Speed Results and Subgroup Analysis

Seven studies assessed gait speed (GS). There were 913 and 938 participants in the exercise intervention and control groups, respectively. Overall, exercise combined with nutritional intervention had a significant effect on gait speed metrics (MD: 0.1 (95% CI: 0.09, 0.11), *p* < 0.00001) with low between-study heterogeneity (*I*^2^ = 34%). Therefore, a fixed-effects model was used for statistical analysis. Mean difference (MD) was used as an effect size indicator. Studies were analyzed in subgroups based on the following four factors, body mass index, age, myasthenia gravis diagnosis, and living environment, as shown in Figure 9.

For subgroup comparisons in GS, BMI-Normal included five studies and BMI-Normal included two studies. The subgroup analyses are shown in Figure 9. In BMI-Abnormal, the MD was 0.1 (95% CI: 0.08, 0.12), *I*^2^ = 58%, *p* < 0.00001. This suggests that exercise combined with nutrition had a significant ameliorative effect on the BMI-Abnormal group; however, there was a high degree of heterogeneity among the studies. In the BMI-Normal group, the MD was 0.12 (95% CI: 0.04, 0.19), *I*^2^ = 0%, *p* = 0.002, which suggests that the intervention of exercise combined with nutrition had a significant ameliorative effect on the BMI-Normal group.

In the residential setting subgroup comparisons, five studies involved independently housed older adults and two studies involved older adults in care settings, as shown in the subgroup analyses in Figure 9. In the independent living subgroup, the MD was 0.1 (95% CI: 0.08, 0.12), *I*^2^ = 58%, *p* < 0.00001). This suggests that the combination of exercise and nutritional interventions had a significant improvement effect in independently housed older adults; however, there was a high degree of heterogeneity across studies. In the subgroup of older adults in care settings, the MD was 0.12 (95% CI: −0.03, 0.26), *I*^2^ = 0%, *p* = 0.11. This finding suggests that the combination of exercise and nutritional interventions did not have a significant ameliorative effect on older adults in care settings.

In the comparison of diagnostic subgroups, two studies included suspected patients and five studies included confirmed patients; subgroup analyses are shown in Figure 9. In the suspected subgroup, the MD was 0.10 (95% CI: 0.08, 0.12), *I*^2^ = 80%, and *p* < 0.00001. This finding suggests that the effect of exercise combined with nutritional interventions was significant; however, there was a high degree of heterogeneity among studies. In the confirmatory subgroup, the MD was 0.14 (95% CI: 0.08, 0.2), *I*^2^ = 0%, *p* < 0.00001. This finding suggests a significant interventional effect of exercise in combination with nutrition.

Subsequently, a publication bias test was performed using REVIEW MANAGER 5.4.1. As shown in Figure 10, the studies were symmetrically distributed; therefore, it can be judged that there was no publication bias. 

### 3.7. Five-Times Sit-to-Stand Results and Subgroup Analysis

Four studies involved five standing tests (five STS tests). In total, 116 and 117 individuals were included in the exercise intervention and control groups, respectively. Overall, exercise combined with nutritional interventions had a significant effect on the five STS indicators (MD: −1.71 [95% CI: −2.32, −1, 1]; *p* < 0.00001), and the heterogeneity across studies was low (*I*^2^ = 10%). Therefore, a fixed-effects model was used for statistical analysis. The mean difference (MD) was used as an indicator of effect size. The studies were analyzed in subgroups based on the following four factors, body mass index, age, myasthenia gravis diagnosis, and living environment, as shown in Figure 11.

For subgroup comparisons of BMI, normal BMI was included in two studies and normal BMI was included in two studies. Subgroup analyses are shown in Figure 11. In BMI-Abnormal, the MD was −1.79 (95% CI: −2.96, −0.62), *I*^2^ = 0%, *p* = 0.003. This indicates that exercise combined with nutrition had a significant ameliorative effect on the BMI-Abnormal group, and there was no heterogeneity among the studies. In the BMI-Normal group, the MD was −1.48 (95% CI: −2.57, to −0.21), *I*^2^ = 71%, *p* = 0.02, which suggests that the intervention of exercise combined with nutrition had a significant ameliorative effect on the BMI-Normal group with a high degree of heterogeneity between studies.

In the residential setting subgroup comparisons, three studies involved independently housed older adults and one study involved older adults in care settings, as shown in the subgroup analyses in Figure 11. In the independently housed subgroup, the MD was −1.63 (95% CI: −2.63, −0.63), *I*^2^ = 44%, *p* = 0.001, which suggests that a combination of exercise and nutritional interventions had a significant ameliorative effect in independently housed older adults, and that there was a low degree of heterogeneity among the studies. In the subgroup of older adults in care settings, the MD was −2.18 (95% CI: −3.88, −0.48), *p* < 0.00001). This finding suggests that exercise combined with nutritional intervention has a significant ameliorative effect on older adults in care settings.

Subsequently, a publication bias test was conducted using REVIEW MANAGER 5.4.1. As shown in Figure 12, the studies were symmetrically distributed; therefore, it can be judged that there was no publication bias. 

## 4. Discussion

This study showed that exercise combined with nutritional intervention demonstrated positive efficacy in improving multiple core outcome indicators in patients with sarcopenia, including functional indicators such as the SMI, HS, GS, and 5-STS, which fully verified the comprehensive effect of the combined interventions in terms of overall muscular health and physical function [30]. However, the intervention effects showed some variability under different population characteristics, suggesting that future clinical practice should pay full attention to individual characteristics to achieve the precision and individualization of intervention strategies.

The substantial heterogeneity observed in several outcome indicators was explored further. Potential sources of heterogeneity include variations in intervention protocols (e.g., the duration, intensity, and mode of exercise or nutrition) [30,35], differences in assessment tools (such as dynamometers for handgrip strength), and heterogeneity in participant characteristics (e.g., age, baseline functional status, and care settings) [51,52]. These methodological and contextual differences may partially account for the inconsistency in the effect sizes. Nonetheless, the sensitivity analyses confirmed the stability of our findings, lending support to the reliability of the overall conclusions.

In response to the observed variability, we expanded the discussion on the clinical implications of the primary outcome measures, including handgrip strength (HS), skeletal muscle index (SMI), gait speed (GS), and the five-times sit-to-stand test (5STS). First, HS is widely recognized as a simple and reliable proxy for the overall muscle strength and physical function in older adults [53,54]. It has been incorporated into the core diagnostic criteria for sarcopenia by international guidelines, such as the EWGSOP2. Low grip strength is strongly associated with an increased risk of falls, hospitalization, disability, and mortality. Therefore, improvements in HS represent a meaningful clinical outcome with direct implications for patient care. Second, the SMI reflects structural aspects of muscle atrophy and is considered a fundamental indicator of sarcopenia. SMI enhancement suggests a potential reversal of degenerative muscle loss and may reflect deeper metabolic and physiological improvements [55]. Third, the GS and 5STS serve as functional markers of mobility and lower extremity strength. Both are predictive of fall risk, independence in daily living, and the overall quality of life [56,57]. Notably, improved GS is linked to sustained autonomy in older adults, whereas enhanced 5STS performance indicates better coordination, neuromuscular function, and lower limb power.

Following this, we elaborated on the clinical feasibility of implementing combined exercise and nutritional interventions across various settings such as community-dwelling older adults, outpatient rehabilitation, and long-term care facilities [58,59]. Special attention was given to their relevance in high-risk subpopulations, such as those with abnormal BMI or early signs of functional decline. These findings support the potential use of targeted interventions to deliver tangible health benefits in real-world clinical and public health contexts.

Several factors may explain the lack of significant effects in certain subgroups. First, baseline functional differences may have introduced a ceiling effect, particularly in individuals who already had relatively preserved muscle strength or cognitive performance before the intervention. In such cases, the limited room for improvement may have obscured the potential benefits of the intervention [60,61]. Second, insufficient intervention intensity and dosage may have attenuated the observed outcomes. In some of the included studies, exercise regimens were of low intensity or short duration (e.g., <8 weeks) and nutritional supplementation failed to meet the recommended protein intake levels (<1.2 g/kg/day) [62,63]. These limitations may be especially pronounced in individuals with an elevated BMI or metabolic disorders, where higher or more individualized dosages are often necessary to produce meaningful effects. Additionally, older adults aged >75 years or those with chronic conditions may exhibit slower physiological adaptations, reduced exercise capacity, or impaired nutrient absorption, which can diminish the impact of interventions. Previous research has also shown that frail older individuals tend to have attenuated responses to training [64]. Moreover, mismatches between intervention type and participant characteristics could have contributed to variability in outcomes. For instance, low-intensity interventions may be inadequate for robust individuals, whereas high-intensity programs may be unsuitable or unsafe for frail populations. This underscores the importance of tailoring intervention strategies according to individual profiles. We detail these considerations in the Section 4 and recommend that future randomized controlled trials incorporate baseline assessments to guide adaptive intervention intensities [65].

To enhance the clinical interpretability of our findings, we have incorporated a discussion of the minimal clinically important differences (MCIDs) for key outcome measures based on the existing literature. For example, a change in handgrip strength (HS) of ≥5.0 kg has been identified as clinically meaningful, while an increase in gait speed (GS) of ≥0.1 m/s is associated with significant improvements in daily functional mobility [66]. In our analysis, the effect sizes reported in multiple studies exceeded these thresholds, suggesting that combined interventions may yield tangible clinical benefits. For outcomes that did not reach the MCID, we cautioned readers to interpret the statistical significance by considering their limited practical relevance.

In addition to the overall effectiveness, we further analyzed the subgroup responses based on BMI, age, sarcopenia diagnosis, and living environment. In terms of the SMI and GS, the effect of the intervention was particularly significant in individuals with an abnormal BMI, including underweight and overweight/obese individuals. This result may be related to the fact that people with an abnormal BMI have a higher risk burden in terms of muscle metabolism, degree of fat infiltration, and level of inflammation and are therefore more likely to respond to exercise and nutrition interventions [63]. Particularly, in overweight and obese patients, resistance training can help improve fat-infiltrated sarcopenic obesity, whereas appropriate nutritional interventions can regulate energy balance and protein synthesis, thereby promoting muscle mass recovery [67]. In addition, in the subgroup of patients with confirmed sarcopenic obesity, the combined intervention also showed good improvement, suggesting that this strategy has clinical value in a population with a clear diagnosis of the disease.

In terms of HS indices, significant improvements were observed only in the under-65 age group. A possible explanation for this is that the 60–65 year age group is more responsive to the intervention because of their better relative retention of exercise capacity, neuromuscular responses, and nutrient uptake and metabolism. In contrast, there are limitations to grip strength improvement in older age groups above 65 years due to muscle attenuation, declining physiological reserve, and a higher proportion of comorbid chronic diseases. In addition, this conclusion is based on limited research data and needs to be validated in future high-quality clinical trials [68].

With regard to living environment, the intervention showed limited efficacy among individuals residing in nursing facilities. This may be attributed to factors such as reduced self-care ability, low adherence to exercise, inconsistent nutritional intake, and variability in caregivers’ expertise [58,69]. Practical challenges, including insufficient rehabilitation guidance and inadequate monitoring, can compromise intervention fidelity. Moreover, cognitive and psychological factors such as depression or social isolation may lower motivation and engagement in institutionalized older adults. These findings highlight the need for more individualized and integrated intervention models in nursing home settings that incorporate regular assessment, multidisciplinary support, and digital tools [70] to improve adherence and outcomes in this vulnerable population.

Further subgroup analyses revealed several sources of heterogeneity in intervention effects across different population characteristics. In participants with an abnormal body mass index (BMI), particularly those who were underweight, combined exercise and nutritional interventions significantly improved muscle function and physical performance [71]. This improvement may be attributed to the rapid stimulation of muscle protein synthesis through nutritional supplementation and the correction of energy deficits. In contrast, individuals with a normal BMI exhibited relatively smaller improvements, likely because of their better baseline muscle mass and functional status. Some studies have suggested that short intervention durations and suboptimal supplementation dosages may limit the effectiveness of this group.

In the analysis of the Five-Times Sit-to-Stand Test (5STS), significant improvements were observed among individuals with an abnormal BMI and those living in community settings (*p* < 0.001). However, no statistical difference was noted in a study by Polo-Ferrero et al. (*p* = 0.127). For some participants with a normal BMI, pre-intervention 5STS scores were already within the normal or high-functioning ranges, leaving limited room for post-intervention gains. These findings underscore the importance of considering baseline functional capacity when interpreting effect sizes and clinical relevance.

Differences were also observed between older adults residing in community and institutional care environments. Community-dwelling individuals generally maintain higher levels of independence and may respond favorably to enhancement-oriented interventions. In contrast, those in nursing homes often present with greater functional impairment, medical comorbidities, and lower adherence levels. For this group, interventions may need to focus more on maintaining function and preventing further decline, rather than enhancement.

Taken together, these results suggest that individual characteristics, such as BMI, baseline functional status, and living environment, can significantly influence the magnitude of the benefits derived from combined exercise and nutritional interventions. Tailored strategies that account for these differences are needed to maximize the efficacy across diverse populations.

In conclusion, exercise combined with nutritional intervention has definite efficacy in sarcopenia, especially in people with an abnormal BMI, those over 65 years of age, those diagnosed with sarcopenia, and those in care settings. However, the effect of the intervention varied in different subgroups, which may be influenced by a variety of factors such as population characteristics, living environment, adherence, and intervention period and intensity. Therefore, future intervention protocols should fully consider individual differences, and higher-quality, standardized clinical studies should be conducted to clarify the optimal modes and dosages of combined interventions in different populations.

It is also important to acknowledge the heterogeneity of responses across subgroups and studies. Patients with an abnormal BMI, a confirmed diagnosis of sarcopenia, or a younger age appear to benefit more from combined exercise and nutritional interventions, whereas those of advanced age or residing in nursing care facilities may require more tailored strategies and improved management approaches. To support more precise and effective sarcopenia interventions, future high-quality, well-designed randomized controlled trials should further investigate how differences in intervention frequency, intensity, duration, and adherence affect outcomes across diverse populations. Furthermore, among the 15 randomized controlled trials included in this review, approximately 80% showed a potential risk of detection bias due to the unclear blinding of outcome assessments, and approximately 13% had incomplete outcome reporting. These methodological limitations suggest that the current findings should be interpreted with caution and that future research should place greater emphasis on rigorous trial design and transparent reporting.

## 5. Limitations

This study had several limitations that warrant consideration. First, the overall sample size and methodological quality of the included randomized controlled trials were limited. The preliminary nature of the findings highlights the need for future studies to expand sample sizes and improve the study quality to strengthen the evidence base. In addition, this review did not perform a sensitivity analysis, which may have affected the robustness of the conclusions.

Significant heterogeneity was found in all 15 included studies. This variation stems partly from the inconsistencies in the diagnostic criteria for sarcopenia. For instance, most European studies employed the EWGSOP2 criteria, which prioritize low muscle strength as the core diagnostic indicator, whereas Asian studies commonly adopted the AWGS 2019 criteria, which incorporate region-specific cutoff values for muscle mass and function. These differing diagnostic frameworks may lead to discrepancies in sarcopenia identification, participant selection, baseline severity, and response to the intervention, ultimately affecting the comparability and generalizability of the results. This underscores the importance of harmonizing diagnostic definitions and acknowledging regional diagnostic standards in future analyses.

The inclusion of the English-language literature may have resulted in language bias. Relevant studies in other languages—especially those from regions with higher prevalence of sarcopenia and active aging research, such as East Asia and South America—may thus have been excluded, limiting the comprehensiveness of the review and potentially leading to an over-representation of the English-language context. Future reviews should consider searching multilingual databases to reduce selection bias and ensure broader representativeness.

Most of the included studies lacked long-term follow-up results, and most of the experimental study periods focused on 12 weeks. This limits our understanding of the sustained effects of combined exercise and nutritional intervention. Owing to the lack of longitudinal data, it is unclear whether the observed improvements in muscle strength, mass, and function are maintained over time and what subsequent effects may be induced.

Given these limitations, future research should prioritize the following directions. High-quality RCTs with larger and more diverse samples, including individuals with multimorbidity and varied cultural and socioeconomic backgrounds, are needed. Intervention protocols should incorporate stratified designs based on baseline function, BMI, age, and living arrangements to enable tailored exercise and nutritional prescriptions. Moreover, long-term studies are essential for assessing the sustainability, adherence, and cost-effectiveness of interventions in real-world clinical and community settings. Standardizing outcome measures and reporting methods will further enhance comparability and allow for a more robust meta-analytical synthesis in future research.

## 6. Conclusions

This study supports and proves that integrated nutrition and exercise improves diseases in the elderly. For the HS indicator, only those under 65 years of age had a significant effect. In the SMI indicator, the intervention effect was significant for body mass index in those greater than 65 years of age. In the GS indicator, the intervention effect was significant for all subgroups except care settings. For the five STS indicators, the intervention effect was significant, regardless of whether the BMI was abnormal.

Notably, the clinical significance of improvements in handgrip strength and gait speed was supported by existing MCID thresholds, reinforcing the practical value of the findings. However, limitations in diagnostic consistency, short intervention durations, potential detection bias, and the lack of long-term follow-up highlight the need for caution in interpretation.

Future high-quality, stratified, and long-term randomized controlled trials are warranted to further refine intervention strategies for diverse older populations and ensure sustainable and personalized sarcopenia management.

## Figures and Tables

**Figure 1 nutrients-17-02342-f001:**
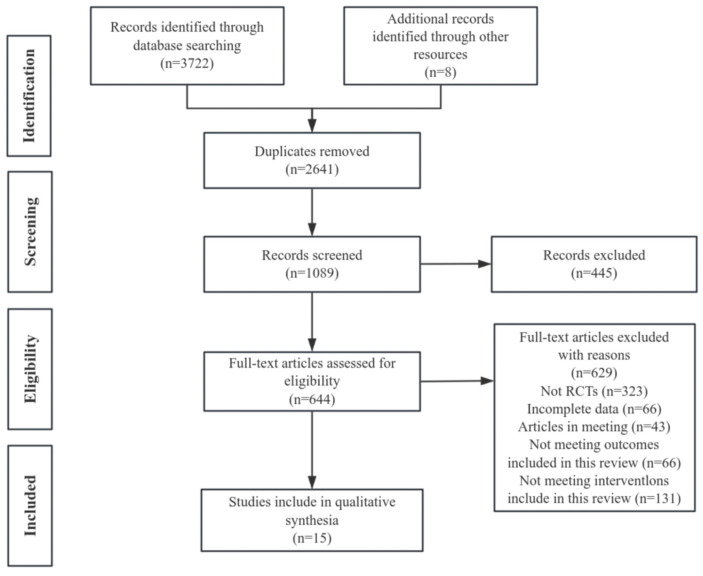
PRISMA study flowchart.

**Figure 2 nutrients-17-02342-f002:**
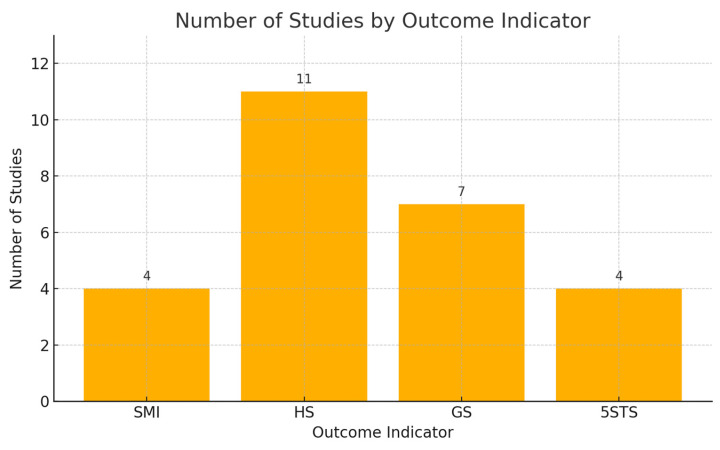
Number of included studies corresponding to each primary outcome indicator. SMI = skeletal muscle index, HS = hand grip strength, GS = gait speed, 5STS = five-times sit-to-stand test.

**Figure 3 nutrients-17-02342-f003:**
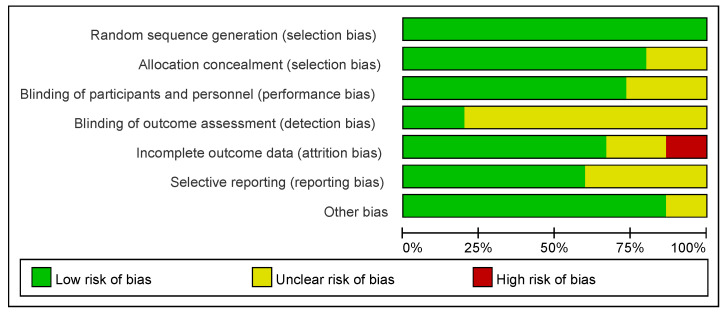
A risk of bias graph showing the percentage of each risk of bias assessment by the review authors for all included studies.

**Figure 4 nutrients-17-02342-f004:**
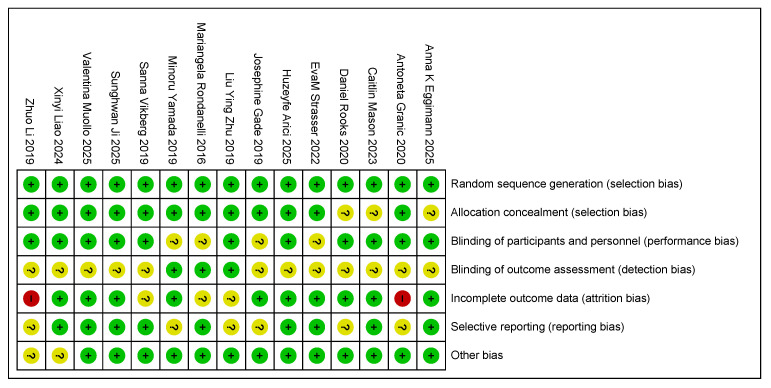
The authors reviewed the assessment of each risk of bias item in each study included in the risk of bias summary [35,36,37,38,39,40,41,42,43,44,45,46,47,48,49]. Red: risk exists; yellow: risk uncertain; green: no risk.

**Figure 5 nutrients-17-02342-f005:**
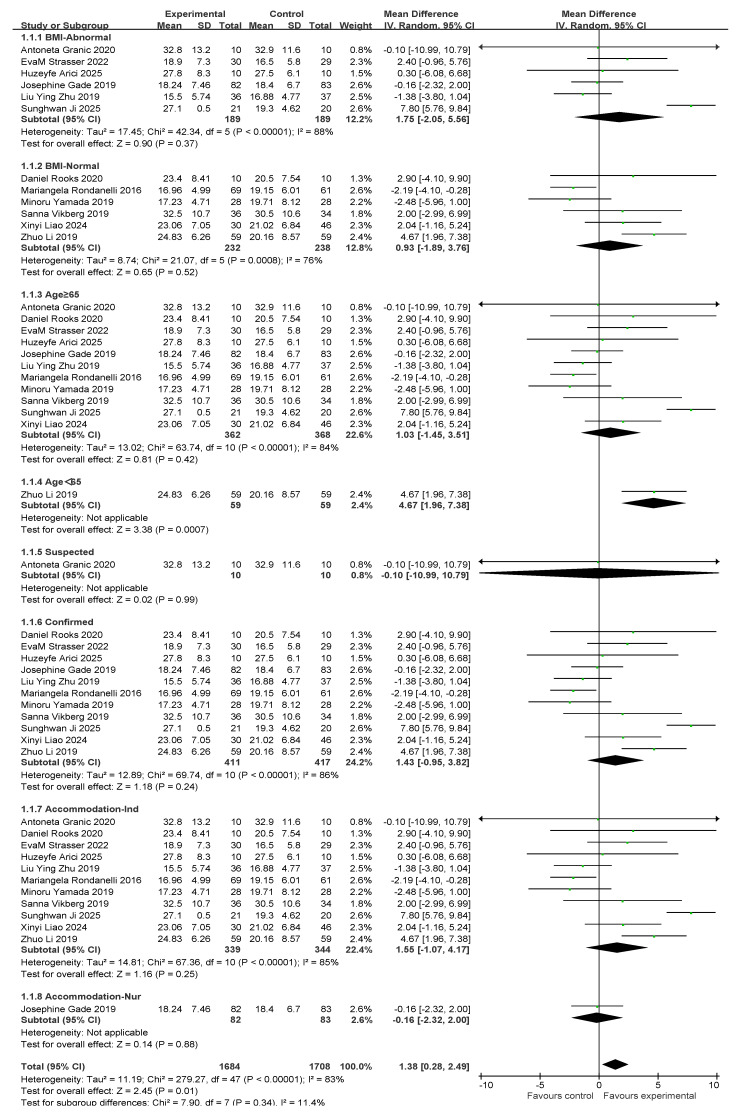
A forest plot of the meta-analysis of the effect of exercise combined with nutrition on HS metrics of sarcopenia interventions in older adults [35,36,37,38,39,40,41,44,46,47,48,49].

**Figure 6 nutrients-17-02342-f006:**
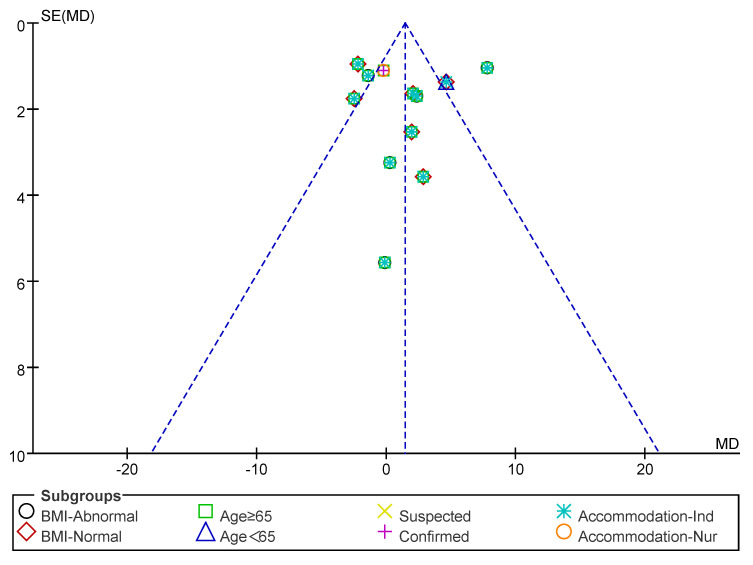
Funnel plot of HS indicators.

**Figure 7 nutrients-17-02342-f007:**
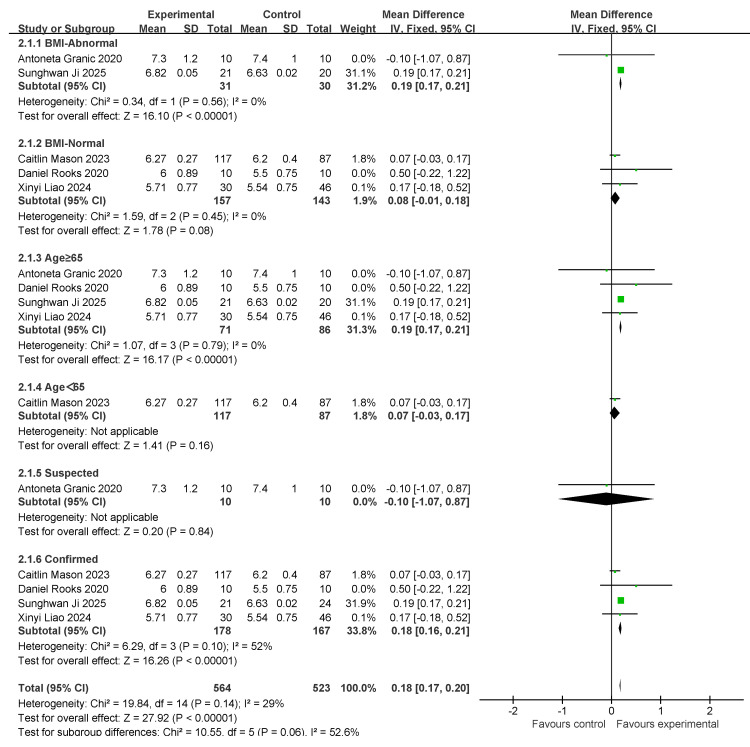
A forest plot of the meta-analysis of the effect of exercise combined with nutrition on SMI metrics of an intervention for sarcopenia in older adults [37,39,41,42,48].

**Figure 8 nutrients-17-02342-f008:**
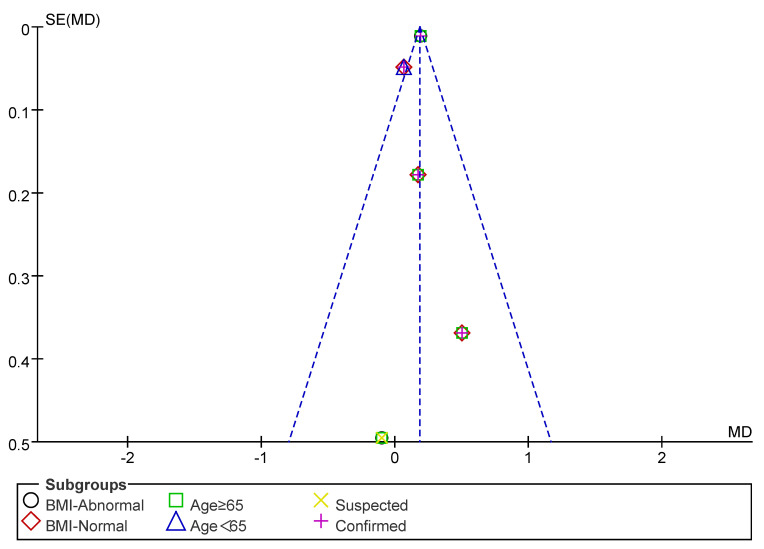
Funnel plot of SMI indicators.

**Figure 9 nutrients-17-02342-f009:**
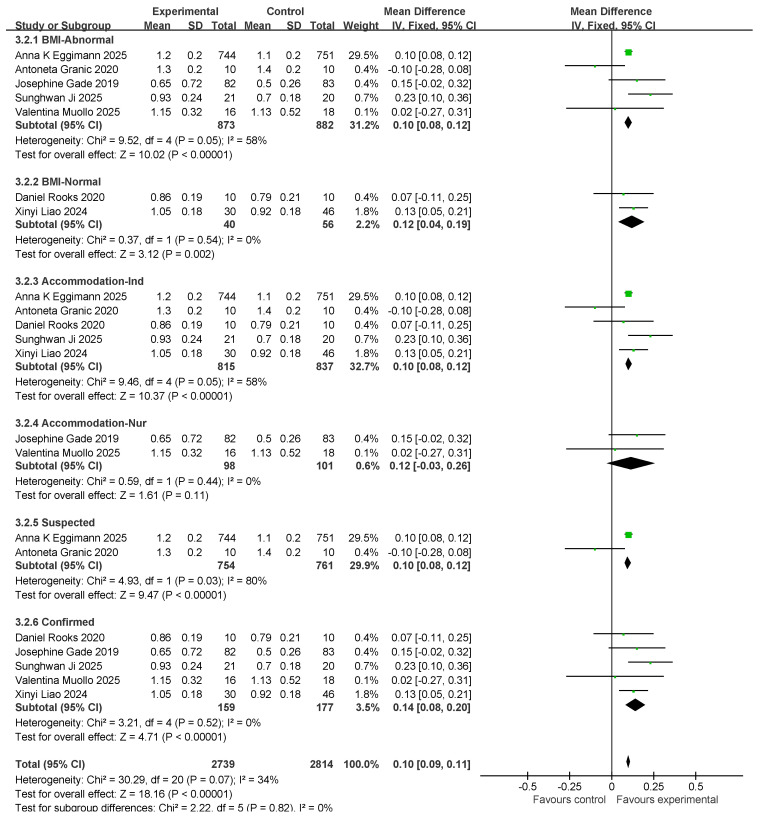
Forest plot of meta-analysis of effect of exercise combined with nutrition on GS metrics in intervention for sarcopenia in older adults [37,39,41,43,45,48,49].

**Figure 10 nutrients-17-02342-f010:**
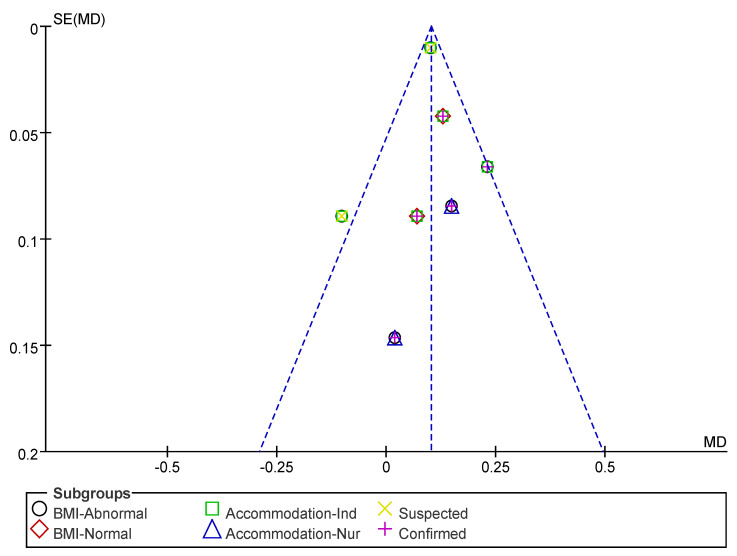
Funnel plot of GS indicators.

**Figure 11 nutrients-17-02342-f011:**
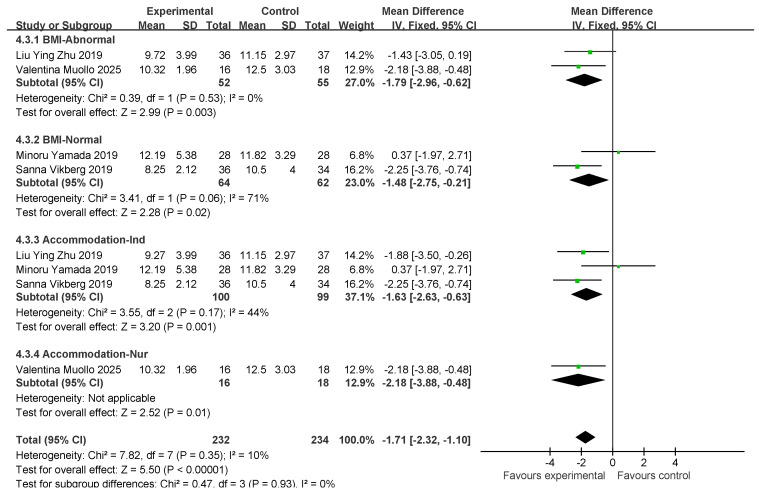
Forest plot of meta-analysis of effect of exercise combined with nutrition on 5 STS metrics in intervention for sarcopenia in older adults [35,38,43,47].

**Figure 12 nutrients-17-02342-f012:**
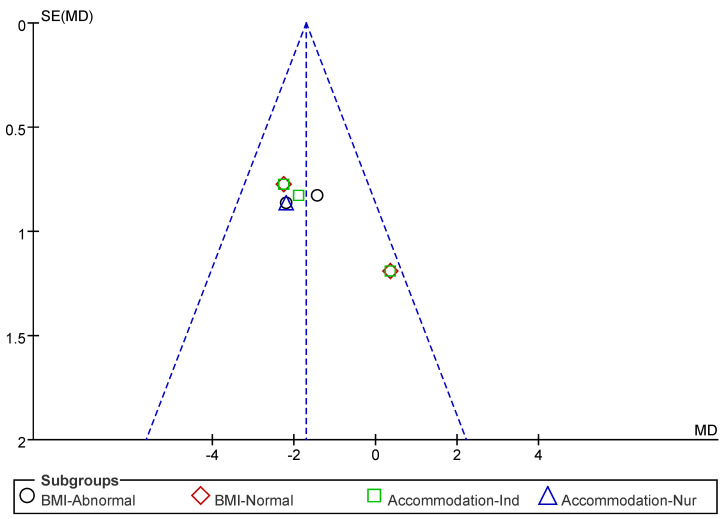
Funnel plot of 5 STS indicators.

**Table 1 nutrients-17-02342-t001:** Characteristics of included articles.

Author (Year)	Country	Population	Diagnostic Criteria Used	Age (Mean + SD)	Total (Male/Female)	Exercise Intervention	Nutrition Intervention	Control	Outcome
Liu-Ying Zhu (2019) [35]	China	Sarcopenia	AWGS 2019	T: 74.8 ± 6.9 C: 72.2 ± 6.6	T: 36/7/29 C: 37/8/29	RT and AE Length of Intervention: 24 weeks Freq: 3 times a week Duration: 60 min	N: protein + vitamin Length of Intervention: 12 weeks Freq: 2 times a day Dose: protein (8.61 g), vitamin D (130 IU)	RCT	HS 5 STS
Zhuo Li (2019) [36]	China	Sarcopenia	AWGS 2019	T: 71.52 ± 5.28 C: 72.91± 6.29	T: 59/22/37 C: 59/12/37	RT & Walking Length of Intervention: 12 weeks Freq: 3 times a week Duration: 30 min	N: protein + vitamin Length of Intervention: 12 weeks Freq: 2 times a day Dose: protein powder (10 g) EPA (300 mg), DHA (200 mg), and vitamin D3 (250 IU)	RCT	GS
Sanna Vikberg (2019) [47]	Sweden	Sarcopenia	EWGSOP2	T: 70.9 ± 0.28 C: 70.9 ± 0.29	T: 36/16/20 C: 34/16/18	RT Length of Intervention: 10 weeks Freq: 3 times a week Duration: 30 min	N: protein Length of Intervention: 10 weeks Freq: 2 times a day Dose: 250 mL liquid, 21 g protein, 1.5 g fat (week 1–7 of the intervention) or 10 g carbohydrates, 30 g protein, and 1.5 g fat (week 8–10 of the intervention)	RCT	HS 5 STS
Huzeyfe Arıcı (2025) [40]	Turkey	Sarcopenia	AWGS 2019	T: 72.6 ± 4.7 C: 71.0 ± 4.5	T: 10/5/5 C: 10/6/4	RT Length of Intervention: 12 weeks Freq: 3 times a week Duration: 60 min	N: protein + VD Length of Intervention: 12 weeks Freq: 12 weeks Dose: protein (1.0–1.2 g/Kg /day)	RCT	HS
Anna K Eggimann (2025) [45]	Switzerland	Sarcopenia	EWGSOP2	T: 74.9 ± 4.4 C: 75.0 ± 4.4	T: 744/273/471 C: 751/278/473	RT +Joint flex Length of Intervention: 3 Years Freq: 3 times a week Duration: 30 min	N: vitamin D + DHA + omega-3s Length of Intervention: 3 years Freq: 1 time/D Dose: vitamin D, 200 UI DHA, omega-3s	RCT	GS
Xinyi Liao (2024) [37]	China	Sarcopenia	AWGS 2019	T: 70.52 ± 3.30 C: 73.21 ± 4.98	T: 30/16/14 C: 46/24/22	RT Length of Intervention: 16 weeks Freq: 5 times a week Duration: 45–60 min	Length of Intervention: 16 weeks Freq: 1 times a day Dose: energy 185 kcal, protein 24.2 g (including plant oligopeptide 11 g, casein peptide 4 g, branch chain amino acid 5 g), CaHMB2.5 g per day	RCT	SMI HS GS
EvaM Strasser (2022) [46]	Austria	Sarcopenia	EWGSOP2	T: 81.5 ± 7.4 C: 83.5 ± 5.7	T: 30/3/27 C: 29/3/26	RT (plastic band) Length of Intervention: 18 months Freq: 1 time/w (1–12 months) 1 time/w/13–18 months Duration: 60 min	Length of Intervention: 18 months Freq: 1 times/w (1–12 months) 1 times/w/13–18 months. Dose: vitamin D, calcium, vitamin b6, vitamin b12, vitamin C, vitamin E, folic acid, and magnesium.	RCT	HS
Daniel Rooks (2020) [41]	USA	Sarcopenia	EWGSOP2	T: 71.7 ± 3.6 C: 71.7 ± 3.6	T: 10/5/5 C: 10/5/5	RE: RT Length of Intervention: 6 weeks Freq: 2 times/week Duration: 60 min	Length of Intervention: 6 weeks Freq: 1 time/after exercise Dose: 3.6 g fat, 3.4 g protein, and 4.7 g carbohydrate per 100 g. And habitual diet	RCT	HS
Minoru Yamada (2019) [38]	Japan	Sarcopenia	AWGS 2019	T: 84.9 ± 5.6 C: 83.9 ± 5.7	T: 28/8/20 C: 28/13/15	RT Length of Intervention: 12 weeks Freq: 2 times/weeks Duration: 30 min	Length of Intervention: 12 weeks Freq: 2 times/weeks Dose: vitamin D	RCT	HS 5STS
Mariangela Rondanelli (2016) [44]	Italy	Sarcopenia	EWGSOP2	T: 80.77 ± 6.29 C: 80.21 ± 8.54	T: 69/29/42 C: 61/24/39	RT Length of Intervention: 12 weeks Freq: 5 times/w/12 Duration: 30 min	Length of Intervention: 12 weeks Freq: 2 times/D/week Dose: vitamin D	RCT	HS
Caitlin Mason (2023) [42]	USA	Sarcopenia	EWGSOP2	T: 57.9 ± 5.0 C: 58.0 ± 5.0	T: 117/88/29 C: 87/65/23	RT + ARE Length of Intervention: 12 months Freq: 5 times/week Duration: 45 min	Length of Intervention: 12 months Freq: 5 times/week Dose: vitamin D	RCT	SMI
Sunghwan Ji (2025) [39]	Korea	Sarcopenia	AWGS 2019	T: 77.86 ± 4.46 C: 78.24 ± 4.47	T: 21/10/11 C: 20/10/11	RT Length of Intervention: 12 weeks Freq: 40 h/week Duration: 45 min	Length of Intervention: 12 weeks Freq: middle of the diet Dose: 13 g/protein, 1.4 fat, 8 carb	RCT	SMI HS GS
Valentina Muollo (2025) [43]	Italy	Sarcopenia	EWGSOP2	T: 65.75 ± 3.94 C: 66.67 ± 3.85	T: 16//8/8 C: 18/9/9	RT Length of Intervention: 12 W Freq: 3/D/W Duration: 60 min	Length of Intervention: 12 W Freq: 2/D/W Dose: EAA + 22 g of protein/day	RCT	GS 5 STS
Antoneta Granic (2020) [48]	UK	Sarcopenia	EWGSOP2	T: 72.2 ± 4.1 C: 70.8 ± 4.0	T: 10/6/4 C: 10/7/3	RT Length of Intervention: 6 weeks Freq: ≥2 times/week Duration:60 min	Length of Intervention: 6 weeks Freq: 1/D Dose: milk 1000 mL, 20 g of protein/day	RCT	GS, HS. SMI
Josepine Gade (2019) [49]	Denmark	Sarcopenia	EWGSOP2	T: 84.2 ± 6.3 C: 85.3 ± 6.2	T: 82/50/32 C: 83/50/30	RT Length of Intervention: Freq: ≥2 times/week Duration: 60 min	Length of Intervention: 12 W Freq: 1/D Dose: milk 1000 mL, 10 g of protein/day	RCT	HS GS

Notes: T: treatment group; C: control group; RT: resistance training; RE: resistance exercise; AE: aerobic training; CON: control group; SMI: skeletal muscle index; AE: aerobic training; CON: control group; SMI: skeletal muscle index; HS: grip strength; GS: gait speed; five tests: five-times sit-to-stand test; EAA: essential amino acid; N: nutrition; W: week; D: day.

## Data Availability

The original contributions presented in the study are included in the article, further inquiries can be directed to the corresponding author.

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
