# Peer review of "Exercise and Nutrition for Sarcopenia: A Systematic Review and Meta-Analysis with Subgroup Analysis by Population Characteristics"

_nutrients, 2025, doi:10.3390/nu17142342_

Round 1
Reviewer 1 Report
Comments and Suggestions for Authors
Thanks for the possibility of evaluating this review of review.
In a synthesis effort, my suggestions for authors are the following:
Introduction
This section is very long from my point of view and oversaturated with epidemiological and physiological data. This can make the purpose of the study no longer clear enough.
Materials and Methods
Exclusion from the eligibility criteria of patients with frequently encountered diseases, such as high blood pressure or liver disease, affects the applicability of study conclusions in the real clinical context. This restrictive selection considerably reduces the capacity to generalize the results, given that the targeted elder population is often characterized by multiple comorbidities.
Results
I recommend the authors to clarify/verify or correct the high heterogeneity in some analyzes (eg HS-83%) is not sufficiently addressed by appropriate techniques (meta-regression or influence analysis).
Also, the graphs (eg: Figure 4-7) are useful, but the associated text is dense, with many values ​​and p-randies, which makes the pursuit for the non-specialist reader.
Discusion
Although this chapter covers the variability between subgroups, I think a deeper analysis of the possible biological mechanisms behind the response differences is lacking.
CONCLUSIONS AND BIBLIOGRAPHY
No comments
Author Response
Kindly refer to the attached document for more details.

Reviewer 2 Report
Comments and Suggestions for Authors
Dear Editor,
Thank you for inviting me to review the manuscript titled "Systematic review and meta-analysis of exercise combined with nutritional interventions for the treatment of sarcopenia: Population characteristics and intervention strategies based on randomized controlled trials to assess effectiveness." This study investigated the effectiveness of combined exercise and nutritional interventions for sarcopenia treatment across different population characteristics. The main findings revealed significant improvements in multiple sarcopenia indicators, with effects varying based on population characteristics such as age, BMI, and living environment.
The current systematic review and meta-analysis addresses a substantial clinical topic pertaining to the treatment of sarcopenia. The methodology used is mostly appropriate, and the results provide valuable insights into population-specific responses to multimodal treatment strategies. However, several elements require improvement to increase the scientific rigor and clinical relevance of the paper.
General comments
- Introduction: The introduction provides adequate background information on sarcopenia and combined interventions but could be enhanced by a more specific rationale for examining population-specific effects and a clearer delineation of research gaps that the proposed study aims to fill.
- Methods: The approach seems generally sound; however, it lacks important details regarding study selection criteria, data extraction procedures, and statistical analysis strategies used. While the PROSPERO registration is noted, the protocol details are insufficient.
- Results: While comprehensive, the results section would benefit from a better presentation of the heterogeneity test and a more logical inclusion of subgroup analyses. The forest plots are informative but would be enhanced by the addition of more statistical measures.
- Discussion: The discussion addresses the main findings but requires deeper critical evaluation of the clinical significance of observed differences and more thorough consideration of study limitations.
Specific Comments
Title and Abstract
- The title is too long and can be shortened without losing its importance.
- The abstract adequately summarizes the research; however, it should include exact effect sizes aligning with the significant results.
- Consider restructuring the abstract to better highlight the novel aspects of population-specific analyses
Introduction
- The rationale for population-specific subgroup analyses needs stronger justification
- Include a clearer statement of the research hypothesis and the expected findings.
- The gap in existing literature regarding population characteristics should be more explicitly stated
- Consider adding a brief paragraph on the clinical significance of the measured outcomes (HS, SMI, GS, 5STS)
Methodology
- The PICOS framework is properly used, but some of the criteria need clarification.
- The exclusion of participants with "disease, hypertension, hematologic disorders, and liver disease" seems overly restrictive and may limit generalizability
- Provide more specific details about the sarcopenia diagnostic criteria accepted across studies
- The search strategy appears to be comprehensive; however, it would be better to include the entire search terms within the main manuscript rather than simply referring to an appendix.
- Consider whether grey literature searches were adequately performed
- The final date for the search, March 31, 2025, appears to be in the future; please confirm.
- More details needed on the data extraction form design and pilot testing
- The evaluation of risk of bias reports the participation of "Two pairs of authors"; however, authorship attribution is unclear.
- Consider providing inter-rater reliability statistics for both data extraction and risk of bias assessment
- The cut-offs for heterogeneity (I² > 50% for random effects) are considered appropriate, but they do require justification.
- Subgroup analysis criteria needs to be predefined and justified.
- Consider addressing potential publication bias through funnel plot analysis and statistical tests
Results
- Table 1 is comprehensive but would be better organized with clearer categorization of the different kinds of interventions.
- The participant demographics show good diversity but age ranges could be better described
- Consider adding a flow diagram showing the number of studies contributing to each outcome
- The forest plots are well-constructed but would benefit from confidence interval reporting in the text
- Effect sizes should be interpreted in a clinical context, for example, minimum clinically important differences.
- The substantial heterogeneity displayed (I² = 83% for grip strength) requires further rigorous examination.
- Perform sensitivity analyses to examine the effect of study quality on the findings.
- The subgroup analyses are valuable but need stronger statistical testing for subgroup differences
- Some subgroups have very few studies (e.g., suspected vs. confirmed sarcopenia), limiting reliability
- Justification for the chosen subgroupings needs to be strengthened.
Discussion
- The discussion addresses main findings but needs deeper consideration of clinical significance
- The explanation for age-related differences in grip strength response needs stronger evidence base
- Consider addressing why some interventions showed no effect in certain populations
- The practical implications for different populations need more detailed discussion
- Consider adding specific recommendations for intervention design based on population characteristics
- Address the cost-effectiveness implications of population-specific interventions
Limitations and Future Research Directions
- The limitations section does not adequately expand and needs more development.
- Investigate the main differences recognized among different assessments.
- Discuss the potential impact of different sarcopenia diagnostic criteria across studies
- Consider the limitation of English-language only studies
- Provide more specific recommendations about future research emphases.
- Recognize the methodological innovations required by foundational research.
- Reflect on options for uniformity of outcome measures.
In conclusion, this systematic review and meta-analysis answers an important clinical question and provides critical insights into the responses of different populations to combined exercise and dietary interventions to alleviate sarcopenia. The strength of the study is indicated by its comprehensive analysis of various population parameters, which has significant practical significance for clinical use.
The manuscript, however, requires major revision to correct methodological flaws, improve statistical detail, and better interpret the clinical meaning of the findings. The considerable heterogeneity seen in several analyses demands further rigorous examination, and the clinical significance of the differences found merits more contextual explanation. After appropriate modifications to address these issues, this manuscript has the promise of making a substantive contribution to the current literature on how sarcopenia should be treated. I recommend that significant changes are required before this manuscript is considered for publication.
Best Regards,
The reviewer
Author Response

(The authors gave the same response as above.)

Reviewer 3 Report
Comments and Suggestions for Authors
Interesting idea of ​​this study, my recommendations are the following:
I recommend rewriting the first sentence, use the word sarcopenia twice.
Line 20 - I recommend mentioning descriptively what the acronyms represent, and in the conclusions to use only the acronyms. In the Methods section I recommend mentioning the number of final articles targeted in the analysis.
Lines 80-95 I recommend mentioning the bibliographical sources that support the mentioned statements.
I recommend enlarging Fig. 1, it is a bit unclear.
Table 1, column 1 I recommend renaming: Authors, and mentioning the names of the authors to be complete, e.g. Rooks D et al, and mentioning the number of the bibliographical source in parentheses.
Fig 4 I recommend enlarging, it is unclear. Idem fig.6.
I recommend that the P value be written in lowercase, according to the editing rules.
Lines 482-488 are recommendations, I recommend deleting or moving to the Practical Implications of the Study section.
Lines 498-502 repeat the idea, I recommend deleting. There is no discussion on this idea.
Lines 515-517 recommend mentioning the bibliographical sources referred to.
Lines 525 recommend mentioning the number of the bibliographical source.
Lines 530-538 recommend moving to the Conclusions section, without duplicating information.
I recommend revising and expanding the Discussions section, by making new concrete correlations, based on bibliographical reference sources, between the results of the present study with results from previous studies.
I recommend expanding the limitations.
I recommend that the bibliography be revised according to the editing rules.
Lines 118-119 the mentioned appendix is ​​not found in the attached document, I recommend clarifications.
Author Response
Kindly refer to the attached document for the revised version of the manuscript.

Round 2
Reviewer 1 Report
Comments and Suggestions for Authors
The authors understood the recommendations and made the respective changes accordingly.
Reviewer 3 Report
Comments and Suggestions for Authors
no comments